# Transfer Learning for Sequences via Learning to Collocate

**Wanyun Cui**[§] **Guangyu Zheng**[‡] **Zhiqiang Shen**[¶] **Sihang Jiang**[‡] **Wei Wang**[‡]

cui.wanyun@sufe.edu.cn, {simonzheng96, zhiqiangshen0214, tedjiangfdu}@gmail.com
weiwang1@fudan.edu.cn

[§]Shanghai University of Finance and Economics
[‡]Shanghai Key Laboratory of Data Science, Fudan University
[¶]Shanghai Key Laboratory of Intelligent Information Processing, Fudan University

## Abstract

Transfer learning aims to solve the data sparsity for a target domain by applying information of the source domain. Given a sequence (e.g. a natural language sentence), the transfer learning, usually enabled by recurrent neural network (RNN), represents the sequential information transfer. RNN uses a chain of repeating cells to model the sequence data. However, previous studies of neural network based transfer learning simply represents the whole sentence by a single vector, which is unfeasible for seq2seq and sequence labeling. Meanwhile, such layer-wise transfer learning mechanisms lose the fine-grained cell-level information from the source domain.

In this paper, we proposed the aligned recurrent transfer, ART, to achieve cell-level information transfer. ART is under the pre-training framework. Each cell attentively accepts transferred information from a set of positions in the source domain. Therefore, ART learns the cross-domain word collocations in a more flexible way. We conducted extensive experiments on both sequence labeling tasks (POS tagging, NER) and sentence classification (sentiment analysis). ART outperforms the state-of-the-arts over all experiments.

## 1 Introduction

Most previous NLP studies focus on open domain tasks. But due to the variety and ambiguity of natural language (Glorot et al., 2011; Song et al., 2011), models for one domain usually incur more errors when adapting to another domain. This is even worse for neural networks since embedding-based neural network models usually suffer from overfitting (Peng et al., 2015). While existing NLP models are usually trained by the open domain, they suffer from severe performance degeneration when adapting to specific domains. This motivates us to train specific models for specific domains.

The key issue of training a specific domain is the insufficiency of labeled data. Transfer learning is one promising way to solve the insufficiency (Jiang & Zhai, 2007). Existing studies (Daumé III, 2007; Jiang & Zhai, 2007) have shown that (1) NLP models in different domains still share many common features (e.g. common vocabularies, similar word semantics, similar sentence syntaxes), and (2) the corpus of the open domain is usually much richer than that of a specific domain.

Our transfer learning model is under the pre-training framework. We first pre-train the model for the source domain. Then we fine-tune the model for the target domain. Recently, some pre-trained models (e.g. BERT (Devlin et al., 2018), ELMo (Peters et al., 2018), GPT-2 (Radford et al., 2019)) successfully learns general knowledge for text. The difference is that these models use a large scale and domain-independent corpus for pre-training. In this paper, we use a small scale but domain-dependent corpus as the source domain for pre-training. We argue that, for the pre-training corpus, the domain relevance will overcome the disadvantage of limited scale.

Most previous transfer learning approaches (Li et al., 2018; Ganin et al., 2016) only transfer information across the whole layers. This causes the information loss from cells in the source domain. ''Layer-wise transfer learning'' indicates that the approach represents the whole sentence by a single

vector. So the transfer mechanism is only applied to the vector. We highlight the effectiveness of *precisely capturing and transferring information of each cell from the source domain* in two cases. First, in seq2seq (e.g. machine translation) or sequence labeling (e.g. POS tagging) tasks, all cells directly affect the results. So layer-wise information transfer is unfeasible for these tasks. Second, even for the sentence classification, cells in the source domain provide more fine-grained information to understand the target domain. For example, in figure 1, parameters for "hate" are insufficiently trained. The model transfers the state of "hate" from the source domain to understand it better.

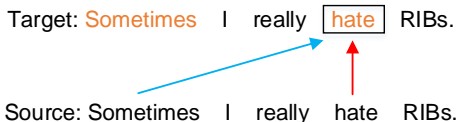

**Figure 1:** Motivation of ART. The orange words "sometimes" and "hate" are with insufficiently trained parameters. The red line indicates the information transfer from the corresponding position. The blue line indicates the information transfer from a collocated word.

Besides transferring the corresponding position's information, the transfer learning algorithm captures the **cross-domain long-term dependency**. Two words that have a strong dependency on each other can have a long gap between them. Being in the insufficiently trained target domain, a word needs to represent its precise meaning by incorporating the information from its collocated words. Here "collocate" indicates that a word's semantics can have long-term dependency on other words. To understand a word in the target domain, we need to precisely represent its collocated words from the source domain. We learn the collocated words via the attention mechanism (Bahdanau et al., 2015). For example, in figure 1, "hate" is modified by the adverb "sometimes", which implies the act of hating is not serious. But the "sometimes" in the target domain is trained insufficiently. We need to transfer the semantics of "sometimes" in the source domain to understand the implication. Therefore, we need to carefully align word collocations between the source domain and the target domain to represent the long-term dependency.

In this paper, we proposed ART (aligned recurrent transfer), a novel transfer learning mechanism, to transfer cell-level information by learning to collocate cross-domain words. ART allows the **cell-level information transfer** by directly extending each RNN cell. ART incorporates the hidden state representation corresponding to the same position and a function of the hidden states for all words weighted by their attention scores.

**Cell-Level Recurrent Transfer** ART extends each recurrent cell by taking the states from the source domain as an extra input. While traditional layer-wise transfer learning approaches discard states of the intermediate cells, ART uses cell-level information transfer, which means each cell is affected by the transferred information. For example, in figure 1, the state of "hate" in the target domain is affected by "sometimes" and "hate" in the source domain. Thus ART transfers more fine-grained information.

**Learn to Collocate and Transfer** For each word in the target domain, ART learns to incorporate two types of information from the source domain: (a) the hidden state corresponding to the same word, and (b) the hidden states for all words in the sequence. Information (b) enables ART to capture the cross-domain long-term dependency. ART learns to incorporate information (b) based on the attention scores (Bahdanau et al., 2015) of all words from the source domain. Before learning to transfer, we first pre-train the neural network of the source domain. Therefore ART is able to leverage the pre-trained information from the source domain.

## 2   ALIGNED RECURRENT TRANSFER

In this section, we elaborate the general architecture of ART. We will show that, ART precisely learns to collocate words from the source domain and to transfer their cell-level information for the target domain.

**Architecture**  The source domain and the target domain share an RNN layer, from which the common information is transferred. We pre-train the neural network of the source domain. Therefore the shared RNN layer represents the semantics of the source domain. The target domain has an additional RNN layer. Each cell in it accepts transferred information through the shared RNN layer. Such

information consists of (1) the information of the same word in the source domain (the red edge in figure 2); and (2) the information of all its collocated words (the blue edges in figure 2). ART uses attention (Bahdanau et al., 2015) to decide the weights of all candidate collocations. The RNN cell controls the weights between (1) and (2) by an update gate.

Figure 2 shows the architecture of ART. The yellow box contains the neural network for the source domain, which is a classical RNN. The green box contains the neural network for the target domain. $S_i$ and $T_i$ are cells for the source domain and target domain, respectively. $T_i$ takes $x_i$ as the input, which is usually a word embedding. The two neural networks overlap each other. The source domain's neural network transfers information through the overlapping modules. We will describe the details of the architecture below. Note that although ART is only deployed over RNNs in this paper, its attentive transfer mechanism is easy to be deploy over other structures (e.g. Transformer (Vaswani et al., 2017))

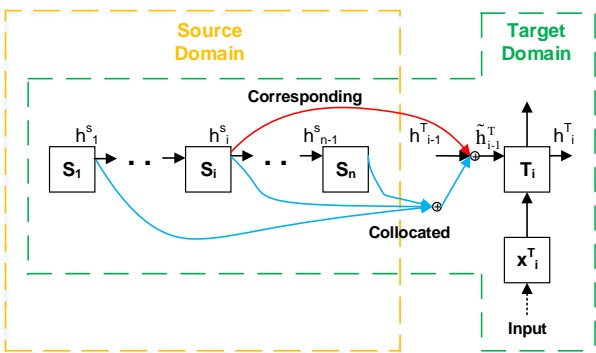

**Figure 2:** ART architecture.

**RNN for the Source Domain** The neural network of the source domain consists of a standard RNN. Each cell $S_i$ captures information from the previous time step $h_{i-1}^S$, computes and passes the information $h_i^S$ to the next time step. More formally, we define the performance of $S_i$ as:

$$h_i^S = RNN(h_{i-1}^S, x_i^S; \theta_S) \tag{1}$$

where $\theta_S$ is the parameter (recurrent weight matrix).

**Information Transfer for the Target Domain** Each RNN cell in the target domain leverages the transferred information from the source domain. Different from the source domain, the $i$-th hidden state in the target domain $h_i^T$ is computed by:

$$h_i^T = RNN(\widetilde{h_{i-1}^T}, x_i^T; \theta_T) \tag{2}$$

where $\widetilde{h_{i-1}^T}$ contains the information passed from the previous time step in the target domain ($h_{i-1}^T$), and the transferred information from the source domain ($\psi_i$). We compute it by:

$$\widetilde{h_{i-1}^T} = f(h_{i-1}^T, \psi_i | \theta_f) \tag{3}$$

where $\theta_f$ is the parameter for $f$.

Note that both domains use the same RNN function with different parameters ($\theta_S$ and $\theta_T$). Intuitively, we always want to transfer the common information across domains. And we think it's easier to represent common and shareable information with an identical network structure.

**Learn to Collocate and Transfer** We compute $\psi_i$ by aligning its collocations in the source domain. We consider two kinds of alignments: (1) The alignment from the corresponding position. This makes sense since the corresponding position has the corresponding information of the source domain. (2) The alignments from all collocated words of the source domain. This alignment is used to represent the long-term dependency across domains. We use a "concentrate gate" $u_i$ to control the ratio between the corresponding position and collocated positions. We compute $\psi_i$ by:

$$\psi_i = (1 - u_i) \circ \pi_i + u_i \circ h_i^S \tag{4}$$

where

$$u_i = \delta(W_u h_i^S + C_u \pi_i) \tag{5}$$

$\pi_i$ denotes the transferred information from collocated words. $\circ$ denotes the element-wise multiplication. $W_u$ and $C_u$ are parameter matrices.

In order to compute $\pi_i$, we use attention (Bahdanau et al., 2015) to incorporate information of all candidate positions in the sequence from the source domain. We denote the strength of the collocation intensity to position $j$ in the source domain as $\alpha_{ij}$. We merge all information of the source domain by a weighted sum according to the collocation intensity. More specifically, we define $\pi_i$ as:

$$\pi_i = \sum_{j=1}^{n} \alpha_{ij} h_j^S \tag{6}$$

where

$$\alpha_{ij} = \frac{\exp(a(h_{i-1}^T, h_j^S))}{\sum_{j'=1}^{n} \exp(a(h_{i-1}^T, h_{j'}^S))} \tag{7}$$

$a(h_i^T, h_j^S)$ denotes the collocation intensity between the $i$-th cell in the target domain and the $j$-th cell in the source domain. The model needs to be evaluated $O(n^2)$ times for each sentence, due to the enumeration of $n$ indexes for the source domain and $n$ indexes for the target domain. Here $n$ denotes the sentence length. By following Bahdanau et al. (2015), we use a single-layer perception:

$$a(h_i^T, h_j^S) = v_a^\top \tanh(W_a h_i^T + U_a h_j^S) \tag{8}$$

where $W_a$ and $U_a$ are the parameter matrices. Since $U_a h_j^S$ does not depend on $i$, we can pre-compute it to reduce the computation cost.

**Update New State** To compute $f$, we use an update gate $z_i$ to determine how much of the source domain's information $\widetilde{\psi}_i$ should be transferred. $\widetilde{\psi}_i$ is computed by merging the original input $x_i$, the previous cell's hidden state $h_{i-1}^T$ and the transferred information $\psi_i$. We use a reset gate $r_i$ to determine how much of $h_{i-1}^T$ should be reset to zero for $\widetilde{\psi}_i$. More specifically, we define $f$ as:

$$f(h_i^T, \psi_i) = (1 - z_i) \circ h_{i-1}^T + z_i \circ \widetilde{\psi}_i \tag{9}$$

where

$$\begin{aligned}
\widetilde{\psi}_i &= \tanh(W_\psi x_i + U_\psi[r_i \circ h_{i-1}^T] + C_\psi \psi_i) \\
z_i &= \delta(W_z x_i + U_z h_{i-1}^T + C_z \psi_i) \\
r_i &= \delta(W_r x_i + U_r h_{i-1}^T + C_r \psi_i)
\end{aligned} \tag{10}$$

Here these $W, U, C$ are parameter matrices.

**Model Training:** We first pre-train the parameters of the source domain by its training samples. Then we fine-tune the pre-trained model with additional layers of the target domain. The fine-tuning uses the training samples of the target domain. All parameters are jointly fine-tuned.

## 3 ART OVER LSTM

In this section, we illustrate how we deploy ART over LSTM. LSTM specifically addresses the issue of learning long-term dependency in RNN. Instead of using one hidden state for the sequential memory, each LSTM cell has two hidden states for the long-term and short-term memory. So the ART adaptation needs to separately represent information for the long-term memory and short-term memory.

The source domain of ART over LSTM uses a standard LSTM layer. The computation of the $t$-th cell is precisely specified as follows:

$$\begin{bmatrix} \widetilde{c_t^S} \\ o_t^S \\ i_t^S \\ f_t^S \end{bmatrix} = \begin{bmatrix} \tanh \\ \sigma \\ \sigma \\ \sigma \end{bmatrix} T_{A,b}^S \begin{bmatrix} x_t^S \\ h_{t-1}^S \end{bmatrix} \tag{11}$$

$$c_t^S = \widetilde{c_t^S} \circ i_t^S + c_{t-1}^S \circ f_t^S \tag{12}$$

$$h_t^S = o_t^S \circ \tanh(c_t^S) \tag{13}$$

Here $h_t^S$ and $c_t^S$ denote the short-term memory and long-term memory, respectively.

In the target domain, we separately incorporate the short-term and long-term memory from the source domain. More formally, we compute the $t$-the cell in the target domain by:

$$\begin{bmatrix} \widetilde{c_i^T} \\ o_t^T \\ i_t^T \\ f_t^T \end{bmatrix} = \begin{bmatrix} \tanh \\ \sigma \\ \sigma \\ \sigma \end{bmatrix} T_{A,b}^T \begin{bmatrix} x_t^T \\ f(h_{i-1}^T, \psi_{hi}|\theta_{fh}) \end{bmatrix} \tag{14}$$

$$c_t^T = \widetilde{c_t^T} \circ i_t^T + f(c_{i-1}^T, \psi_{ci}|\theta_{fc}) \circ f_t^T \tag{15}$$

$$h_t^T = o_t^T \circ \tanh(c_t^T) \tag{16}$$

where $f(h_{i-1}^T, \psi_{hi}|\theta_{fh})$ and $f(c_{i-1}^T, \psi_{ci}|\theta_{fc})$ are computed by Eq (6) with parameters $\theta_{fh}$ and $\theta_{fc}$, respectively. $\psi_{hi}$ and $\psi_{ci}$ are the transferred information from the short-term memory ($h_1^S \cdots h_n^S$ in Eq. (12)) and the long-term memory ($c_1^S \cdots c_n^S$ in Eq. (13)), respectively.

**Bidirectional Network** We use the bidirectional architecture to reach all words' information for each cell. The backward neural network accepts the $x_i (i = 1 \ldots n)$ in reverse order. We compute the final output of the ART over LSTM by concatenating the states from the forward neural network and the backward neural network for each cell.

## 4 EXPERIMENTS

We evaluate our proposed approach over sentence classification (sentiment analysis) and sequence labeling task (POS tagging and NER).

### 4.1 SETUP

All the experiments run over a computer with Intel Core i7 4.0GHz CPU, 32GB RAM, and a GeForce GTX 1080 Ti GPU.

**Network Structure:** We use a very simple network structure. The neural network consists of an embedding layer, an ART layer as described in section 3, and a task-specific output layer for the prediction. We will elaborate the output layer and the loss function in each of the tasks below. We use 100d GloVe vectors (Pennington et al., 2014) as the initialization for ART and all its ablations.

**Competitor Models** We compare ART with the following ablations.

- **LSTM** (no transfer learning): It uses a standard LSTM without transfer learning. It is only trained by the data of the target domain.
- **LSTM-u** It uses a standard LSTM and is trained by the union data of the source domain and the target domain.
- **LSTM-s** It uses a standard LSTM and is trained only by the data of the source domain. Then parameters are used to predicting outputs of samples in the target domain.
- **Layer-Wise Transfer (LWT)** (no cell-level information transfer): It consists of a layer-wise transfer learning neural network. More specifically, only the last cell of the RNN layer transfers information. This cell works as in ART. LWT only works for sentence classification. We use LWT to verify the effectiveness of the cell-level information transfer.
- **Corresponding Cell Transfer (CCT)** (no collocation information transfer): It only transfers information from the corresponding position of each cell. We use CCT to verify the effectiveness of collocating and transferring from the source domain.

We also compare ART with state-of-the-art transfer learning algorithms. For sequence labeling, we compare with hierarchical recurrent networks (HRN) (Yang et al., 2017) and FLORS (Schnabel & Schütze, 2014; Yin, 2015). For sentence classification, we compare with DANN (Ganin et al., 2016), DAmSDA (Ganin et al., 2016), AMN (Li et al., 2017), and HATN (Li et al., 2018). Note that FLORS, DANN, DAmSDA, AMN and HATN use labeled samples of the source domain and unlabeled samples of both the source domain and the target domain for training. Instead, ART and HRN use labeled samples of both domains.

## 4.2 SENTENCE CLASSIFICATION: SENTIMENT ANALYSIS

**Datasets:** We use the Amazon review dataset (Blitzer et al., 2007), which has been widely used for cross-domain sentence classification. It contains reviews for four domains: *books, dvd, electronics, kitchen*. Each review is either positive or negative. We list the detailed statistics of the dataset in Table 1. We use the training data and development data from both domains for training and validating. And we use the testing data of the target domain for testing.

Table 1: Statistics of the Amazon review dataset. *%Neg.* denotes the ration of the negative samples. *Avg. L* denotes the average length of each review. *Vocab.* denotes the vocabulary size.

|  | Train | Dev. | Test | % Neg. | Avg. L | Vocab. |
|---|---|---|---|---|---|---|
| Books | 1400 | 200 | 400 | 50% | 159 | 62k |
| Electronics | 1398 | 200 | 400 | 50% | 101 | 30k |
| DVD | 1400 | 200 | 400 | 50% | 173 | 69k |
| Kitchen | 1400 | 200 | 400 | 50% | 89 | 28k |

**Model Details:** To adapt ART to sentence classification, we use a max pooling layer to merge the states of different words. Then we use a perception and a sigmoid function to score the probability of the given sentence being positive. We use binary cross entropy as the loss function. The dimension of each LSTM is set to 100. We use the Adam (Kingma & Ba, 2015) optimizer. We use a dropout probability of 0.5 on the max pooling layer.

Table 2: Classification accuracy on the Amazon review dataset.

| Source | Target | LSTM | LSTM-u | LSTM-s | CCT | LWT | DANN | DAmSDA | AMN | HATN | ART |
|---|---|---|---|---|---|---|---|---|---|---|---|
| Books | DVD | 0.695 | 0.770 | 0.718 | 0.730 | 0.784 | 0.725 | 0.755 | 0.818 | 0.813 | **0.870** |
| Books | Elec. | 0.733 | 0.805 | 0.678 | 0.768 | 0.763 | 0.690 | 0.760 | 0.820 | 0.790 | **0.848** |
| Books | Kitchen | 0.798 | 0.845 | 0.678 | 0.818 | 0.790 | 0.770 | 0.760 | 0.810 | 0.738 | **0.863** |
| DVD | Books | 0.745 | 0.788 | 0.730 | 0.800 | 0.778 | 0.745 | 0.775 | 0.825 | 0.798 | **0.855** |
| DVD | Elec. | 0.733 | 0.788 | 0.663 | 0.775 | 0.785 | 0.745 | 0.800 | 0.810 | 0.805 | **0.845** |
| DVD | Kitchen | 0.798 | 0.823 | 0.708 | 0.815 | 0.785 | 0.780 | 0.775 | 0.830 | 0.765 | **0.853** |
| Elec. | Books | 0.745 | 0.740 | 0.648 | 0.773 | 0.735 | 0.655 | 0.725 | 0.785 | 0.763 | **0.868** |
| Elec. | DVD | 0.695 | 0.753 | 0.648 | 0.768 | 0.723 | 0.720 | 0.695 | 0.780 | 0.788 | **0.855** |
| Elec. | Kitchen | 0.798 | 0.863 | 0.785 | 0.823 | 0.793 | 0.823 | 0.838 | **0.893** | 0.808 | 0.890 |
| Kitchen | Books | 0.745 | 0.760 | 0.653 | 0.803 | 0.755 | 0.645 | 0.755 | 0.798 | 0.740 | **0.845** |
| Kitchen | DVD | 0.695 | 0.758 | 0.678 | 0.750 | 0.748 | 0.715 | 0.775 | 0.805 | 0.738 | **0.858** |
| Kitchen | Elec. | 0.733 | 0.815 | 0.758 | 0.810 | 0.805 | 0.810 | **0.870** | 0.833 | 0.850 | 0.853 |
| Average | | 0.763 | 0.792 | 0.695 | 0.803 | 0.774 | 0.735 | 0.774 | 0.817 | 0.783 | **0.858** |

**Results:** We report the classification accuracy of different models in Table 2. The no-transfer LSTM only performs accuracy of 76.3% on average. ART outperforms it by 9.5%. ART also outperforms its ablations and other competitors. This overall verifies the effectiveness of ART.

**Effectiveness of Cell-Level Transfer** LWT only transfers layer-wise information and performs accuracy of 77.4% on average. But ART and CCT transfer more fine-grained cell-level information. CCT outperforms LWT by 2.9%. ART outperforms LWT by 8.4%. This verifies the effectiveness of the cell-level transfer.

**Effectiveness of Collocation and Transfer** CCT only transfers the corresponding position's information from the source domain. It achieves accuracy of 80.3% on average. ART outperforms CCT by 5.5% on average. ART provides a more flexible way to transfer a set of positions in the source domain and represent the long-term dependency. This verifies the effectiveness of ART in representing long-term dependency by learning to collocate and transfer.

**Minimally Supervised Domain Adaptation** We also evaluate ART when the number of training samples for the target domain is much fewer than that of the source domain. For each target domain in the Amazon review dataset, we combine the training/development data of rest three domains as the source domain. We show the results in Table 3. ART outperforms all the competitors by a large margin. This verifies its effectiveness in the setting of minimally supervised domain adaptation.

Table 3: Classification accuracy with scarce training samples of the target domain.

| Target | LSTM | LSTM-u | LSTM-s | CCT | LWT | HATN | ART |
|--------|------|--------|--------|-----|-----|------|-----|
| Books | 0.745 | 0.813 | 0.800 | 0.848 | 0.808 | 0.820 | **0.895** |
| DVD | 0.695 | 0.748 | 0.795 | 0.870 | 0.770 | 0.828 | **0.875** |
| Electronics | 0.733 | 0.823 | 0.760 | 0.848 | 0.818 | 0.863 | **0.865** |
| Kitchen | 0.798 | 0.840 | 0.795 | 0.860 | 0.840 | 0.833 | **0.870** |
| Average | 0.743 | 0.806 | 0.788 | 0.856 | 0.809 | 0.836 | **0.876** |

## 4.3 SEQUENCE LABELING

We evaluate the effectiveness of ART w.r.t. sequence labeling. We use two typical tasks: POS tagging and NER (named entity recognition). The goal of POS tagging is to assign part-of-speech tags to each word of the given sentence. The goal of NER is to extract and classify the named entity in the given sentence.

**Model Settings:** POS tagging and NER are multi-class labeling tasks. To adapt ART to them, we follow HRN (Yang et al., 2017) and use a CRF layer to compute the tag distribution of each word. We predict the tag with maximized probability for each word. We use categorical cross entropy as the loss function. The dimension of each LSTM cell is set to 300. By following HRN, we use the concatenation of 50d word embeddings and 50d character embeddings as the input of the ART layer. We use 50 1d filters for CNN char embedding, each with a width of 3. We use the Adagrad (Duchi et al., 2011) optimizer. We use a dropout probability of 0.5 on the max pooling layer.

**Dataset:** We use the dataset settings as in Yang et al. (2017). For POS Tagging, we use Penn Treebank (PTB) POS tagging, and a Twitter corpus (Ritter et al., 2011) as different domains. For NER, we use CoNLL 2003 (Tjong Kim Sang & De Meulder, 2003) and Twitter (Ritter et al., 2011) as different domains. Their statistics are shown in Table 4. By following Ritter et al. (2011), we use 10% training samples of Twitter (Twitter/0.1), 1% training samples of Twitter (Twitter/0.01), and 1% training samples of CoNLL (CoNLL/0.01) as the training data for the target domains to simulate a low-resource setting. Note that the label space of these tasks are different. So some baselines (e.g. LSTM-u, LSTM-s) cannot be applied.

Table 4: Dataset statistics.

| Benchmark | Task | # Training Tokens | # Dev Tokens | # Test Tokens |
|-----------|------|-------------------|--------------|---------------|
| PTB | POS Tagging | 912,344 | 131,768 | 129,654 |
| Twitter | POS Tagging | 12,196 | 1,362 | 1,627 |
| CoNLL 2003 | NER | 204,567 | 51,578 | 46,666 |
| Twitter | NER | 36,936 | 4,612 | 4,921 |

Table 5: Performance over POS tagging and NER.

| Task | Source | Target | HRN | FLORS | LSTM | CCT | ART |
|------|--------|--------|-----|-------|------|-----|-----|
| POS Tagging | PTB | Twitter/0.1 | 0.837 | 0.763 | 0.798 | 0.852 | **0.859** |
| POS Tagging | PTB | Twitter/0.01 | 0.647 | | 0.573 | 0.653 | **0.658** |
| NER | CoNLL | Twitter/0.1 | 0.432 | - | 0.210 | 0.434 | **0.450** |
| NER | Twitter | CoNLL/0.01 | - | - | 0.576 | 0.675 | **0.707** |

Table 5 shows the per-word accuracy (for POS tagging) and f1 (for NER) of different models. From the table, we see that the performances of all tasks are improved by ART. For example, when transferring from PTB to Twitter/0.1, ART outperforms HRN by 2.2%. ART performs the best among all competitors in almost all cases. This verifies the effectiveness of ART w.r.t. sequence labeling. Note that FLORS is independent of the target domain. If the training corpus of the target domain is quite rare (Twitter/0.01), FLORS performs better. But with richer training data of the target domain (Twitter/0.1), ART outperforms FLORS.

## 4.4 VISUALIZATION OF THE ALIGNED TRANSFER

ART aligns and transfers information from different positions in the source domain. Intuitively, we use the alignment and attention matrices to represent cross-domain word dependencies. So positions with stronger dependencies will be highlighted during the transfer. We visualize the attention matrix for sentiment analysis to verify this. We show the attention matrices for the short-term memory $h$ and for the long-term memory $c$ in figure 3.

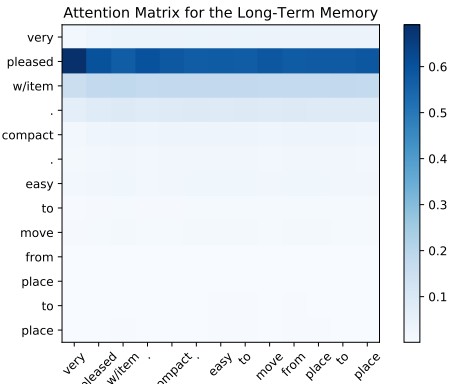 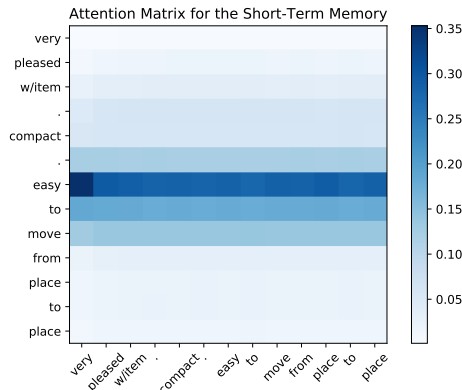

**Figure 3:** Attention matrix visualization. The x-axis and the y-axis denote positions in the target domain and source domain, respectively. Figure (a) shows the attention matrix for the long-term memory in the forward neural network. Figure (b) shows the attention matrix for the short-term memory in the forward neural network.

**Effectiveness of the Cell-Level Transfer** From figure 3, words with stronger emotions have more attentions. For example, the word "pleased" for the long-term memory and "easy to move" for the short-term memory have strong attention for almost all words, which sits well with the intuition of cell-level transfer. The target domain surely wants to accept more information from the meaningful words in the source domain, not from the whole sentence. Notice that $c$ and $h$ have different attentions. Thus the two attention matrices represent discriminative features.

**Effectiveness in Representing the Long-Term Dependency** We found that the attention reflects the long-term dependency. For example, in figure 3 (b), although all words in the target domain are affected by the word "easy", the word "very" has highest attention. This makes sense because "very" is actually the adverb for "easy", although they are not adjacent. ART highlights cross-domain word dependencies and therefore gives a more precise understanding of each word.

## 5 RELATED WORK

**Neural network-based transfer learning** The layer-wise transfer learning approaches (Glorot et al., 2011; Ajakan et al., 2014; Zhou et al., 2016) represent the input sequence by a non-sequential vector. These approaches cannot be applied to seq2seq or sequence labeling tasks. To tackle this problem, algorithms must transfer cell-level information in the neural network (Yang et al., 2017). Some approaches use RNN to represent cell-level information. Ying et al. (2017) trains the RNN layer by domain-independent auxiliary labels. Ziser & Reichart (2018) trains the RNN layer with pivots. However, the semantics of a word can depend on its collocated words. These approaches cannot represent the collocated words. In contrast, ART successfully represents the collocations by attention.

**Pre-trained models** ART uses a pre-trained model from the source domain, and fine-tunes the model with additional layers for the target domain. Recently, pre-trained models with additional layers are shown to be effectiveness for many downstream models (e.g. BERT (Devlin et al., 2018), ELMo (Peters et al., 2018)). As a pre-trained model, ELMo uses bidirectional LSTMs to generate contextual features. Instead, ART uses attention mechanism in RNN that each cell in the target domain directly access information of all cells in the source domain. ART and these pre-trained models have different goals. ART aims at transfer learning for one task in different domains, while BERT and ELMo focus on learning general word representations or sentence representations.

## 6 CONCLUSION

In this paper, we study the problem of transfer learning for sequences. We proposed the ART model to collocate and transfer cell-level information. ART has three advantages: (1) it transfers more fine-grained cell-level information, and thus can be adapted to seq2seq or sequence labeling tasks; (2) it aligns and transfers a set of collocated words in the source sentence to represent the cross domain long-term dependency; (3) it is general and can be applied to different tasks. Besides, ART verified the effectiveness of pre-training models with the limited but relevant training corpus.

ACKNOWLEDGMENTS

Guangyu Zheng and Wei Wang were supported by Shanghai Software and Integrated Circuit Industry Development Project(170512).

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
