# OpenReview forum: "Transfer Learning for Sequences via Learning to Collocate"
_ICLR.cc/2019/Conference_

### Official Review · AnonReviewer2 · 2018-11-01
**Reasonable idea but the technical details are quite unclear**

**Rating:** 6
**Confidence:** 4

**Review:**

This paper presents the following approach to domain adaptation. Train a source domain RNN. While doing inference on the target domain, first you run the source domain RNN on the sequence. Then while running the target domain RNN, set the hidden state at time step i, h^t_i, to be a function 'f' of  h^t_{i-1} and information from source domain \psi_i; \psi_i is computed as a convex combination of the state of the source domain RNN, h^s_{i}, and an attention-weighted average of all the states h^s{1...n}. So in effect, the paper transfers information from each of source domain cells -- the cell at time step i and all the "collocated" cells (collocation being defined in terms of attention). This idea is then extended in a straightforward way to LSTMs as well.

Doing "cell-level" transfer enables more information to be transferred according to the authors, but it comes at a higher computation since we need to do O(n^2) computations for each cell.

The authors show that this beats a variety of baselines for classification tasks (sentiment), and for sequence tagging task (POS tagging over twitter.)

Pros:
1. The idea makes sense and the experimental results show solid

Cons:
1. Some questions around generalization are not clearly answered. E.g. how are the transfer parameters of function 'f' (that controls how much source information is transferred to target) trained? If the function 'f' and the target RNN is trained on target data, why does 'f' not overfit to only selecting information from the target domain? Would something like dropping information from target domain help?

2. Why not also compare with a simple algorithm of transferring parameters from source to target domain? Another simple baseline is to just train final prediction function (softmax or sigmoid) on the concatenated source and target hidden states. Why are these not compared with? Also, including the performance of simple baselines like word2vec/bow is always a good idea, especially on the sentiment data which is very commonly used and widely cited.

3. Experiments: the authors cite the hierarchical attention transfer work of Li et al (https://www.aaai.org/ocs/index.php/AAAI/AAAI18/paper/download/16873/16149) and claim their approach is better, but do not compare with them in the experiments. Why?

Writing:
The writing is quite confusing at places and is the biggest problem with this paper. E.g.

1. The authors use the word "collocated" everywhere, but it is not clear at all what they mean. This makes the introduction quite confusing to understand. I assumed it to mean words in the target sentences that are strongly attended to. Is this correct? However, on page 4, they claim "The model needs to be evaluated O(n^2) times for each sentence pair." -- what is meant by sentence pair here? It almost leads me to think that they consider all source sentence and target sentences? This is quite confusing.

2. The authors keep claiming that "layer-wise transfer learning mechanisms lose the fine-grained cell-level information from the source domain", but it is not clear exactly what do they mean by layer-wise here. Do they mean transferring the information from source cell i to target cell i as it is? In the experiments section on LWT, the authors claim that "More specifically, only the last cell of the RNN layer transfers information. This cell works as in ART. LWT only works for sentence classification." Why is it not possible to train a softmax over both the source hidden state and the target hidden state for POS tagging?

nits:
page 4 line 1: "i'th cell in the source domain" -> "i'th cell in the target domain". "j'th cell in target" -> "j'th cell in sourcE".


Revised: increased score after author response.

---

> ### Author Response · Authors · 2018-11-26
> **Response to AnonReviewer2 [new baselines and clarifications]**
>
> Thank you for your insightful and supportive comments. We have made the following revisions: (1) We added two baselines according to your comments. The results further justify the effectiveness of ART. (2) We clarified “collocate”, “layer-wise transfer learning”, “model training”, and their related issues. We give more details below:
>
> 1. Regarding computational cost:
> The network depth only increases by 2 if we ignore the detailed operations (e.g. gates). One is caused by collocating and transferring. Another one is caused by merging the original input, the previous cell’s hidden state, and the transferred information. So the time cost does not increase much.
>
> 2. Regarding Con1: why does 'f' not overfit to only selecting information from the target domain?
> Your understanding is correct. Function 'f' will overfit to the target domain. All parameters will be jointly fine-tuned by the training samples of the target domain. Nevertheless, the pre-training for the source domain still helps because it provides representations of the source domain. Another recent successful example of using pre-trained models is BERT (Devlin et al., 2018), which also fine-tunes all the parameters to specific tasks.
> And we rewrite the model training part in section 2 to make it clearer.
> “We first pre-train the parameters of the source domain by its training samples. Then we fine-tune the pre-trained model with additional layers of the target domain. The fine-tuning uses the training samples of the target domain. All parameters are jointly fine-tuned.”
>
> Regarding Con2. More simple baselines.
> First, we added a baseline model, LSTM-s, which directly uses parameters from the source domain to the target domain. The results are shown in Table 2. ART outperforms the baseline by a large margin.
>
> Table 2: Classiﬁcation accuracy on the Amazon review dataset.
> Source		Target		LSTM-s	HATN	ART
> Books		DVD		0.718	0.813	0.870
> Books		Electronics	0.678	0.790	0.848
> Books		Kitchen		0.678	0.738	0.863
> DVD		Books		0.730	0.798	0.855
> DVD		Electronics	0.663	0.805	0.845
> DVD		Kitchen		0.708	0.765	0.853
> Electronics	Books		0.648	0.763	0.868
> Electronics	DVD		0.648	0.788	0.855
> Electronics	Kitchen		0.785	0.808	0.890
> Kitchen		Books		0.653	0.740	0.845
> Kitchen		DVD		0.678	0.738	0.858
> Kitchen		Electronics	0.758	0.850	0.853
>         Average				0.695	0.783	0.858
>
> Second, you suggest directly concatenating the hidden states of the source and the target domains. In fact, we already proposed a very similar baseline CCT. The only difference is that CCT uses a gate to merge the two values, instead of concatenation. ART outperforms CCT in all cases.
> Third, we already used 100d GloVe vectors to initialize ART and all its ablations we proposed in this paper. The pre-trained word embeddings are also widely used by its competitors (e.g. AMN and HATN). We have added the description in section 4.
>
>
> Regarding Con 3. Experiments: the hierarchical attention transfer work of Li et al.
> We added the comparison with HATN (Li et al 2018). The results are shown in Table 2 and Table 3. We use the source code and hyper parameters of (Li et al 2018) from the authors’ Github. We changed its labeled training samples from 5600 to 1400 as with ART.
>
> The results are shown in Table 2 above. ART still beats the baseline by a large margin. This verifies its effectiveness.
>
> == Writing ==
> Regarding Writing1.
> First, for the meaning of “collocate”, we added more explanations and take figure 1 as an example in section 1.
> “Here “collocate” indicates that a word's semantics can have long-term dependency on other words. To understand a word in the target domain, we need to precisely represent its collocated words from the source domain. We learn from the collocated words via the attention mechanism. For example, in figure 1, “hate” is modified by the adverb “sometimes”, which implies the act of hating is not serious. But the “sometimes” in the target domain is trained insufficiently. We need to transfer the semantics of “sometimes” in the source domain to understand the implication.”
> Second, to avoid the ambiguity of “sentence pair”, we rewrote the description in the revised version.
> “The model needs to be evaluated O(n^2) times for each sentence, due to the enumeration of n indexes for the source domain and n indexes for the target domain. Here n denotes the sentence length.”
>
> Regarding Writing2.
> “Layer-wise transfer learning” indicates that the approach represents the whole sentence by a single vector. So the transfer mechanism is only applied to the vector. We cannot apply layer-wise transfer learning algorithms to sequence labeling tasks.
> We added the descriptions in section 1.

---

> > ### Comment · AnonReviewer2 · 2018-11-26
> > **Solids results; still on the fence**
> >
> > Thanks for providing the latest set of results. Your experimental results are quite solid and so I am improving my score. However not giving it very high scores because I still feel a little hesitant about the writing quality in this paper. The technical writing is still subpar. E.g.
> > 1) "ART discriminates between information of the corresponding position and that of all positions with collocated words." => you probably want to say "ART incorporates the hidden state representation corresponding to the same position and a function of the hidden states for all other words weighted by their attention scores"
> > 2) "By using the attention mechanism (Bahdanau et al., 2015), we compute the correlation for each word pair" => correlation has a very specific meaning and it makes it confusing if you use here.
> >
> > There are several such examples.

---

> > > ### Author Response · Authors · 2018-11-27
> > > **Writing**
> > >
> > > Thank you for your encouraging comments.
> > > We agree that there is room for writing of the original submission. We have been improving the writing quality. We believe that the latest version is much clearer now.
> > >
> > > We made the following revisions to improve the writing:
> > > 1. We gave more descriptions of how ART works.
> > > i. [Learn to Collocate and Transfer] In section 1, we rewrote paragraph of “learn to collocate and transfer”. We highlighted how ART incorporates two types of information and uses the attention mechanism to capture the long-term cross-domain dependency.
> > > ii. [Architecture] In section 2, we added a paragraph to describe the architecture of ART. We elaborated how it incorporates the information of the source domain from the pre-trained model.
> > > iii. [Model training] In section 2, we rewrote the paragraph of model training. We highlighted the model pre-training procedure and fine-tuning procedure of ART.
> > > 2. We added the interpretations and examples for some confusing notions, such as “level-wise transfer learning”, “cell-level transfer learning”, and “collocate”.
> > > 3. We abandoned or reduced some vague words or phrases, such as “word correlation”, “collocate”. The revised version uses more precise expressions, such as “dependencies between two words”, “incorporate information by their attention score”.
> > > 4. We rewrote the related work section. We compared ART with BERT and ELMo. The latter two approaches also use pre-trained models for downstream tasks.
> > > 5. We fixed some typos.

---

> > > ### Author Response · Authors · 2018-11-29
> > > **Detailed rewritings**
> > >
> > > For your detailed writing advices, we have rewritten the two sentences accordingly.
> > >
> > > 1.	We rewrote the sentence
> > > “ART discriminates between information of the corresponding position and that of all positions with collocated words.”
> > > to
> > > “For each word in the target domain, ART learns to incorporate two types of information from the source domain: (a) the hidden state corresponding to the same word, and (b) the hidden states for all words in the sequence.”
> > >
> > > 2.	We rewrote the sentence
> > > “By using the attention mechanism (Bahdanau et al., 2015), we compute the correlation for each word pair.”
> > > to
> > > “ART learns to incorporate information (b) based on the attention scores (Bahdanau et al., 2015) of all words from the source domain.”
> > >
> > > For more writing improvements, please refer to the previous comment or the paper.

---

### Official Review · AnonReviewer3 · 2018-11-02
**The paper proposed to use RNN/LSTM with collocation alignment as a representation learning method for transfer learning/domain adaptation in NLP.**

**Rating:** 5
**Confidence:** 4

**Review:**

The proposed method is suitable for many NLP tasks, since it can handle the sequence data.

I find it difficult to follow through the model descriptions.  Perhaps a more descriptive figures would make this easier to follow, I feel that the ART model is a very strait forward and it can be easily described in much simpler and less exhausting (sorry for the strong word) way, while there is nothing wrong with being as elaborating as you are, I feel that all those details belong in an appendix.
Can you please explain the exact learning process?
I didn’t fully understand the exact way of collocations, you first train on the source domain and then use the trained source network when training in the target domain with all the collocated words for each training example? I deeply encourage you to improve the model section for future readers.
In contrast to the model section, the related work and the experimental settings sections are very thin.
The experimental setup for the sentiment analysis experiments is quite rare in the transfer learning/domain adaptation landscape, having equal amount of labeled data from both source and target domains is not very realistic in my humble opinion.
More realistic setup is unsupervised domain adaptation (like in DANN and MSDA-DAN papers) or minimally supervised domain adaptation (like you did in your POS and NER experiments).

In addition to the LSTM baseline (which is trained with target data only), I think that LSTM which is trained on both source and target domains data is required for truly understand ART gains – this goes for the POS and NER tasks as well.
The POS and NER experiments can use some additional baselines for further comparison, for example:
http://www.aclweb.org/anthology/Q14-1002
https://hornhehhf.github.io/hangfenghe/papers/14484-66685-1-PB.pdf

I am not sure I understand the “cell level transfer” claim, did you mean that you are the first to apply inner LSTM/RNN cell transfer or that you are the first ones to apply word-level fine grained transfer, the latter has already been done:
https://arxiv.org/pdf/1802.05365.pdf
https://ink.library.smu.edu.sg/cgi/viewcontent.cgi?article=4531&context=sis_research
http://www.aclweb.org/anthology/N18-1112
https://openreview.net/pdf?id=rk9eAFcxg

---

> ### Author Response · Authors · 2018-11-26
> **Response to AnonReviewer3 [new baselines, experimental settings, and clarifications]**
>
> Thank you for your insightful and supportive comments. We have made the following revisions: (1) We added two baselines according to your comments. The results further justify the effectiveness of ART. (2) We added a new experiment for minimally supervised domain adaptation in Table 3. ART still outperforms all the competitors by a large margin. (3) We clarified the ART model and model training process in the revised paper. We will give more details below:
>
> == Writing ==
> 1. High level description of the ART model.
> We have added the following description of ART model in section 2.
> “The source domain and the target domain share an RNN layer, from which the common information is transferred. We pre-train the neural network of the source domain. Therefore the shared RNN layer represents the semantics of the source domain. The target domain has an additional RNN layer. Each cell in it accepts transferred information through the shared RNN layer. Such information consists of (1) the information of the same word in the source domain (the red edge in figure 2); and (2) the information of all its collocated words (the blue edges in figure 2). ART uses attention to decide the weights of all candidate collocations. The RNN cell controls the weights between (1) and (2) by an update gate.”
>
> 2. Model training. We add more details of the model training part in section 2.
> We first pre-train the parameters of the source domain by its training samples. Then we fine-tune the pre-trained model with additional layers of the target domain. The fine-tuning uses the training samples of the target domain. All parameters are jointly fine-tuned.
>
> 3. Related work.
> We have rewritten the related work section. We compare with other cell-level transfer learning approaches and pre-trained models.
>
> == Innovation of cell-level transfer ==
> We agree that some previous transfer learning approaches also consider cell-level transfer. But none of them considers the word collocations. As a pre-trained model, ELMo uses bidirectional LSTMs to generate contextual features. Instead, ART uses attention mechanism in RNN that each cell in the target domain directly access information of all cells in the source domain. We added more details in the related work section.
>
> == Baselines ==
> We added two baselines, LSTM-u and FLORS, according to your comments. LSTM-u uses a standard LSTM and is trained by the union data of the source and the target domain. FLORS is a domain adaptation model for POS tagging (http://www.aclweb.org/anthology/Q14-1002). Their results are shown in Table 2 and Table 5. ART outperforms LSTM-u in almost all settings by a large margin. Note that FLORS is independent of the target domain. If the training corpus of the target domain is quite rare (Twitter/0.01), FLORS performs better. But with richer training data of the target domain (Twitter/0.1), ART outperforms FLORS by a large margin.
>
> Table 2: Classiﬁcation accuracy on the Amazon review dataset.
> Source		Target		LSTM-u	ART
> Books		DVD		0.770 	0.870
> Books		Electronics	0.805 	0.848
> Books		Kitchen		0.845 	0.863
> DVD		Books		0.788 	0.855
> DVD		Electronics	0.788 	0.845
> DVD		Kitchen		0.823 	0.853
> Electronics	Books		0.740 	0.868
> Electronics	DVD		0.753 	0.855
> Electronics	Kitchen		0.863 	0.890
> Kitchen		Books		0.760 	0.845
> Kitchen		DVD		0.758 	0.858
> Kitchen		Electronics	0.815 	0.853
>         Average				0.792 	0.858
>
> Table 5: Performance over POS tagging.
> Task			Source	Target		FLORS	ART
> POS Tagging	        PTB		Twitter/0.1	0.763	0.859
> POS Tagging	        PTB		Twitter/0.01	0.763	0.658
>
> == Experimental settings ==
> Based on your comment, we added a new experiment for minimally supervised domain adaptation in sentence classification. For each target domain in the Amazon review dataset, we combined the training/development data of rest three domains as the source domain. We show the results in Table 3. ART outperforms the competitors by a large margin. This verifies its effectiveness in the setting of minimally supervised domain adaptation.
>
> Table 3: Classification accuracy with scarce training samples of the target domain.
> Target		LSTM	LSTM-u	CCT		LWT	HATN	ART
> Books		0.745 	0.813 	0.848 	0.808 	0.820 	0.895
> DVD		0.695 	0.748 	0.870 	0.770 	0.828 	0.875
> Electronics	0.733 	0.823 	0.848 	0.818 	0.863 	0.865
> Kitchen		0.798 	0.840 	0.860 	0.840 	0.833 	0.870
> Average		0.743 	0.806 	0.856 	0.809 	0.836 	0.876

---

### Official Review · AnonReviewer1 · 2018-11-03
**Good empirical results on transfer learning; writing could be clearer**

**Rating:** 6
**Confidence:** 3

**Review:**

== Quality of results ==
This paper's empirical results are its main strength. They evaluate on a well-known benchmark for transfer learning in text classification (the Amazon reviews dataset of Blitzer et al 2007), and improve by a significant margin over recent state-of-the-art methods. They also evaluate on several sequence tagging tasks and achieve good results.

One weakness of the empirical results is that they do not compare against training a model on the union of the source and target domain. I think this is very important to compare against.

Note: the authors cite a paper in the introduction "Hierarchical Attention Transfer Network for Cross-domain Sentiment
Classification" (Li et al 2018) which also achieves state of the art results on the Amazon reviews dataset, but do not compare against it. At first glance, Li et al 2018 appear to get better results. However, they appear to be training on a larger amount of data for each domain (5600 examples, rather than 1400). It is unclear to me why their evaluation setup is different, but some clarification about this would be helpful.

== Originality ==
A high level description of their approach:
1. Train an RNN encoder ("source domain encoder") on the source domain
2. On the target domain, encode text using the following strategy:
  - First, encode the text using the source domain encoder
  - Then, encode the text using a new encoder (a "target domain encoder") which has the ability to attend over the hidden states of the source domain encoder at each time step of encoding.

They also structure the target domain encoder such that at each time step, it has a bias toward attending to the hidden state in the source encoder at the same position.

This has a similar flavor to greedy layer-wise training and model stacking approaches. In that regard, the idea is not brand new, but feels well-applied in this setting.

== Clarity ==
I felt that the paper could have been written more clearly. The authors set up a comparison between "transfer information across the whole layers" vs "transfer information from each cell" in both the abstract and the intro, but it was unclear what this distinction was referring to until I reached Section 4.1 and saw the definition of Layer-Wise Transfer.

Throughout the abstract and intro, it was also unclear what was meant by "learning to collocate cross domain words". After reading the full approach, I see now that this simply refers to the attention mechanism which attends over the hidden states of the source domain encoder.

== Summary ==
This paper has good empirical results, but I would really like to see a comparison against training a model on the union of the source and target domain. I think superior results against that baseline would increase my rating for this paper.

I think the paper's main weakness is that the abstract and intro are written in a way that is somewhat confusing, due to the use of unconventional terminology that could be replaced with simpler terms.

---

> ### Author Response · Authors · 2018-11-26
> **Response to AnonReviewer1 [new baselines and clarifications]**
>
> Thank you for your insightful and supportive comments. We have made the following revisions: (1) We added two baselines based on your comments. The results further justified the effectiveness of ART. (2) We added the clarification of “layer-wise transfer learning”, “cell-level transfer learning”, and “collocate” in section 1. We will give more details below:
>
> ==Experiments==
> We added two baselines, LSTM-u and HATN, according to your comments. LSTM-u uses a standard LSTM and is trained by the union data of the source domain and the target domain. The HATN model is from the paper "Hierarchical Attention Transfer Network for Cross-domain Sentiment Classification" (Li et al 2018). We use the source code and hyper parameters of (Li et al 2018) from the authors’ Github. We changed its labeled training samples from 5600 to 1400 as with ART.
>
> The results are shown in Table 2. ART still beats the baselines by a large margin. This verifies its effectiveness.
>
> Table 2: Classiﬁcation accuracy on the Amazon review dataset.
> Source		Target		LSTM-u	HATN	ART
> Books		DVD		0.770	0.813	0.870
> Books		Electronics	0.805	0.790	0.848
> Books	    	Kitchen		0.845	0.738	0.863
> DVD 	        Books		0.788	0.798	0.855
> DVD	        Electronics	0.788	0.805	0.845
> DVD	        Kitchen		0.823	0.765	0.853
> Electronics    	Books		0.740	0.763	0.868
> Electronics	DVD		0.753	0.788	0.855
> Electronics	Kitchen		0.863	0.808	0.890
> Kitchen		Books		0.760	0.740	0.845
> Kitchen		DVD		0.758	0.738	0.858
> Kitchen		Electronics	0.815	0.850	0.853
>          Average			0.792	0.783	0.858
>
>
> == Writing==
> We added more detailed explanations and took figure 1 as an example to clarify the confusing parts in section 1.
>
> 1. Layer-wise transfer learning:
> “Layer-wise transfer learning” indicates that the approach represents the whole sentence by a single vector. So the transfer mechanism is only applied to the vector.
>
> 2. Cell-level transfer learning:
> ART uses cell-level information transfer, which means each cell is affected by the transferred information.  For example, in figure 1, the state of “hate” in the target domain is affected by “sometimes” and ”hate” in the source domain.
>
> 3. Collocate:
> We use the term “collocate” to indicate that a word's semantics can have long-term dependency on another word. To understand a word in the target domain, we need to precisely capture and represent its collocated words from the source domain. We learn from the collocated words via the attention mechanism. For example, in figure 1, “hate” is modified by the adverb “sometimes”, which implies the act of hating is not serious. But “sometimes” in the target domain is trained insufficiently. We need to transfer the semantics of “sometimes”.

---

> > ### Public Comment · ~zheng_li4 · 2018-12-22
> > **Poor results about the HATN model**
> >
> > I have got this information from my github issues (https://github.com/hsqmlzno1/HATN/issues/5). I'm the author of the hatn model. I think the author of this paper may have reported inaccurate results about the hatn model in the rebuttal. I have also verified the hatn model in the small-scale setting, the results still remain to be superior, which is largely better than the reported results. So I think the author need to check the experiments and tune the hyper-parameters if the setting has been changed, which could be more promising. Thanks!

---

> > > ### Author Response · Authors · 2018-12-22
> > > **detailed experimental settings for HATN**
> > >
> > > Hi Zheng, let's try to make it clearer and reach an accurate agreement for HATN.
> > >
> > > As described in the response above, we used the source code and keeped hyper parameters in https://github.com/hsqmlzno1/HATN. More specifically, we initialize HATN by 300d skip-gram vectors. The dimensions of the word attention layer and the sentence attention layer are both 300. We train the model with batch_size=50, learning rate=1e-4. We use the same early-stopping policy. Besides, we use the same unlabeled data provided from https://github.com/hsqmlzno1/HATN. These settings are all from your github repository.
> > >
> > > Please provide more details of your implementation and you results. We will consider updating the results in the camera ready version if we find the results change a lot in your settings.

---

> > > > ### Public Comment · ~zheng_li4 · 2018-12-22
> > > > **Missing some points**
> > > >
> > > > Hi Cui,
> > > >
> > > > Thanks for your reply.
> > > >
> > > > I think you may ignore some critical points.
> > > >
> > > > First, the experiment setting may be unfair. According to the description of your paper "We use the training data and development data from both domains for training and validating. And we use the testing data of the target domain for testing", your method should be supervised domain adaptation method. However, you have compared with many unsupervised domain adaptation methods in a supervised setting.
> > > >
> > > > Second, I have mentioned that if the setting has been changed, you should tune the hyper-parameters, not only just use the original setup in a different setting.
> > > >
> > > > You can send me your original raw data split (zlict@connect.ust.hk) such that I can give you updated results in the next few days.

---

> > > > > ### Author Response · Authors · 2018-12-22
> > > > > **Response**
> > > > >
> > > > > Thanks for your concerns.
> > > > >
> > > > > Your first concern is already addressed by AnonReviewer3. Please refer to our response to AnonReviewer3 about minimally supervised domain adaptation.
> > > > >
> > > > > For your second concern, we will try to fine-tune the hyperparameters and see if it changes a lot before the camera ready. We will also release our source code and datasets later.

---

> > > > > > ### Public Comment · ~zheng_li4 · 2018-12-22
> > > > > > **Response**
> > > > > >
> > > > > > I think the first concern has not been addressed by AnonReviewer3. You still use 1400 target domain labeled data for training. Do your think it is minimally supervised domain adaptation in a small-scale setting? You can check some supervised domain methods (e.g., http://aclweb.org/anthology/P18-1233), they only use 50 target domain labeled data.
> > > > > >
> > > > > > I have to say, even without the aid of the combination of rest three domains, the hierarchical attention network (with MLP, not GRU Yang et.al)  can achieve better results based on 1400 target domain labeled data than the reported results in Table 3.

---

### Meta-Review · Area_Chair1 · 2018-12-14

**Confidence:** 4
**Recommendation:** Accept (Poster)

**Metareview:**

This paper presents a method for transferring source information via the hidden states of recurrent networks.  The transfer happens via an attention mechanism that operates between the target and the source.  Results on two tasks are strong.

I found this paper similar in spirit to Hypernetworks (David Ha, Andrew Dai, Quoc V Le, ICLR 2016) since there too there is a dynamic weight generation for network given another network, although this method did not use an attention mechanism.

However, reviewers thought that there is merit in this paper (albeit pointed the authors to other related work) and the empirical results are solid.